# UnDREAM: Bridging Differentiable Rendering and Photorealistic Simulation for End-to-end Adversarial Attacks

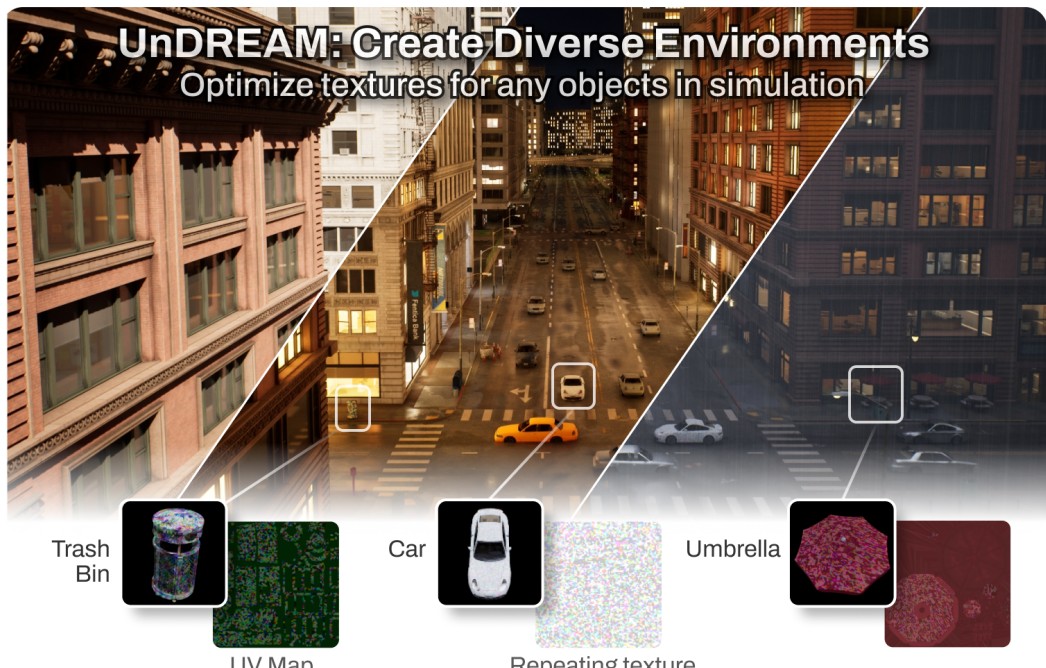

Figure 1: UnDREAM is the first software framework bridging differentiable rendering and photorealistic simulation to enable end-to-end adversarial attacks. Users can create diverse environments by controlling environmental conditions, add and configure custom 3D objects and execute adversarial attacks that faithfully follow threat model.

## Abstract

Deep learning models deployed in safety critical applications like autonomous driving use simulations to test their robustness against adversarial attacks in realistic conditions. However, these simulations are non-differentiable, forcing researchers to create attacks that do not integrate simulation environmental factors, reducing attack success. To address this limitation, we introduce UnDREAM, the first software framework that bridges the gap between photorealistic simulators and differentiable renderers to enable end-to-end optimization of adversarial perturbations on any 3D objects. UnDREAM enables manipulation of the environment by offering complete control over weather, lighting, backgrounds, camera angles, trajectories, and realistic human and object movements, thereby allowing the creation of diverse scenes. We showcase a wide array of distinct physically plausible adversarial objects that UnDREAM enables researchers to swiftly explore in different configurable environments. This combination of photorealistic simulation and differentiable optimization opens new avenues for advancing research of physical adversarial attacks.

## 1 INTRODUCTION

Ensuring the adversarial robustness of vision systems is important, as computer vision is applied in safety-critical domains like autonomous vehicles. For example, adversarial perturbations when put on stop signs can deceive object detection systems, potentially causing them to misclassify or fail to recognize traffic signs altogether (Wei et al., 2024). Adversarial perturbations are typically generated through an iterative optimization process without ever inserting the perturbations into the simulator itself. This begins with an initial pattern, commonly referred to as a "*patch*," alongside a collection of environmental images in which the patch is to be evaluated. This patch is optimized for the desired adversarial task. The optimization process does not account for complex interactions between the patch and the environment, such as lighting variations, or material properties. As a result, when the optimized patch is eventually placed into the simulation environment, its visual appearance often changes dramatically, due to these unmodeled physical factors, leading to reduced adversarial effectiveness, as shown in Fig. 2. This observation aligns with findings from prior work, which report similar degradation in performance under such conditions (Xu et al., 2024). This issue is particularly concerning, given that simulations are intended as a proxy for real world situations (Wei et al., 2024). Consequently, the failure of the attack to remain effective under realistic lighting and material conditions raises significant concerns about its transferability from digital to physical environments (Nesti et al., 2022; Hu et al., 2023).

Figure 2: Existing attack pipelines create adversarial attacks outside of simulation without accounting for interaction with the environment, resulting in attacks that fail when the patches are integrated in the simulation.

**Why do researchers continue to use this sub-optimal pipeline?** A primary reason is that the photorealistic simulation platforms such as Unreal Engine (Engine, 2018) that allow for complex human and object movements, and realistic physics modules for collision and light scattering (Wei et al., 2024), are **all non-differentiable** (Table 1). Researchers have tried inserting patches into the simulation itself and estimating gradients using PyTorch transformations (Xu et al., 2024). However, such approaches restrict the patch to shapes and transformations that can be approximated in two dimensions, limiting their applicability in creating complex 3D adversarial objects dimensions (Nesti et al., 2024). While differentiable renderers are often used in the creation of adversarial textures they lack other attributes to act as a satisfactory substitute, such as simulating movement of objects and an easy to use interface (Hull et al., 2024). This fundamental gap explains why recent adversarial methods use this optimization pipeline or other similar approximation techniques; direct access to gradients within realistic simulation environments is unavailable, preventing end-to-end optimization of textures (Xu et al., 2024). As a result, there is not a unified approach that enables differentiable optimization within 3D simulation, a critical gap in research creating a barrier to research progress (Wei et al., 2024).

We propose UnDREAM: **D**ifferentiable **R**endering for **E**nd-to-end **A**dversarial **M**odeling in Unreal Engine, a novel, practical software framework that bridges the gap between photorealistic 3D simulation environments and differentiable optimization, thereby offering the best of both worlds. Specifically, UnDREAM enables the conversion of 3D objects and textures from Unreal Engine into a fully differentiable system, facilitating end-to-end optimization of adversarial patterns of arbitrarily shaped 3D objects within photorealistic simulation environments. We make the following key contributions:

| Framework | Code/Tool Available | Edit Assets | Custom Assets | Differentiable Textures | Rendering/Tech Stack |
|---|---|---|---|---|---|
| **UnDREAM** (Ours) | ✓ | ✓ | ✓ | ✓ | Unreal 5.6 (Latest) |
| Carla (Dosovitskiy et al., 2017) | ✓ | ✓ | ✓ | ✗ | Unreal 4.26 (2020) |
| AirSim (Shah et al., 2017) | ✓ | ✓ | ✗ | ✗ | Unreal 4.27 (2021) |
| Carla-Gear (Nesti et al., 2024) | ✗ | ✓ | ✗ | ✗ | Unreal 4.26 |
| Carla Drone (Meier et al., 2024a) | ✓ | ✗ | ✗ | ✗ | Unreal 4.26 |
| SkyScenes (Khose et al., 2024) | ✓ | ✓ | ✗ | ✗ | Unreal 4.26 |
| SynDrone (Rizzoli et al., 2023) | ✓ | ✓ | ✗ | ✗ | Unreal 4.26 |
| UrbanScene3D (Lin et al., 2022) | ✓ | ✓ | ✗ | ✗ | Unreal 4.27 |

Table 1: Comparison of UnDREAM with existing simulation frameworks and data generators. UnDREAM is the **only** framework that simultaneously allows editing, environment customization and differentiation of textures on the 3D objects in realistic simulation.

1. **UnDREAM: First Software Framework Bridging Differentiable Rendering and Photorealistic Simulation for End-to-End Adversarial Attacks.** Unlike existing approaches that optimize textures in isolation and later superimpose them (Fig. 2), UnDREAM enables optimization as the texture appears natively in the simulation, preserving lighting, perspective, and physical material interactions. (§ 3.2)

2. **Automatic, Precise 3D Transformation Enabling Faithful Threat Modeling.** By embedding adversarial textures directly into the simulation environment, UnDREAM eliminates the need for calculating the bounds of the adversarial object across This leads to better alignment with realistic threat models and provides a more faithful evaluation of both adversarial attacks and corresponding defenses. (§ 3.1.2)

3. **Open-Source Implementation Enabling New Research Directions.** UnDREAM offers a flexible and scalable interface that allows researchers to optimize adversarial textures on 3D objects of arbitrary shape with minimal code modification (e.g., a single-line change) This flexibility, combined with the realism of high-fidelity simulation, opens the door to novel research directions in physical adversarial attacks (§ 4). UnDREAM's anonymized repository is available at https://anonymous.4open.science/r/UnDREAM and is ready for immediate public release.

## 2 RELATED WORK

### 2.1 SIMULATION IN ADVERSARIAL MACHINE LEARNING

High-fidelity simulators have become a cornerstone for developing and testing autonomous systems (Wu et al., 2020; Zhang et al., 2024; Nesti et al., 2022). These platforms provide a critical bridge between digital adversarial optimization and physical-world evaluation by offering photorealistic graphics, dynamic environments, and configurable sensor suites. Simulators such as CARLA (Dosovitskiy et al., 2017) and Airsim (Shah et al., 2017), both of which are built on Unreal Engine (Engine, 2018), excel at modeling real-world conditions including variable weather, lighting, and camera perspectives - making them ideal for generating diverse and scalable datasets. This capability has led to the creation of numerous synthetic datasets for tasks like semantic segmentation and object detection (Fonder, 2019; Wang et al., 2020; Lin et al., 2022; Rizzoli et al., 2023; Khose et al., 2024; Meier et al., 2024b). These datasets offer enhanced controllability and scalability compared to real-world benchmarks (Nigam et al., 2018; Lyu et al., 2020; Cai et al., 2025), enabling more systematic evaluation of perception models.

However, a fundamental limitation persists across all these simulation frameworks: they are non-differentiable. Table 1 shows a detailed comparison of UnDREAM and existing frameworks, highlighting it as the only framework that simultaneously allows environment customization and differentiation of textures on the 3D objects in realistic simulation While platforms facilitate dataset generation they provide no support for differentiable textures or gradient-based editing in the simulation. Thus, even though these frameworks are valuable during evaluation and data generation, they

are lacking when it comes to the ability to seamlessly fit in the adversarial optimization pipeline. As a result, adversarial studies using these frameworks or datasets typically resort to post-hoc superimposition of adversarial textures onto rendered images. This approach breaks the rendering consistency, causing adversarial objects to appear disharmonious with the scene by failing to account for lighting, shadows, and occlusion (Xu et al., 2024). This limitation not only reduces visual realism but also severely undermines the transferability and effectiveness of such attacks in real-world scenarios (Nesti et al., 2022).

## 2.2 DIFFERENTIABLE RENDERING

Differentiable rendering bridges the gap between traditional computer graphics and gradient-based optimization by enabling gradients to propagate from an image-based loss function back to underlying scene parameters, such as geometry, lighting, and textures (Loper & Black, 2014; Kato et al., 2017; Laine et al., 2020). This capability has proven especially valuable in adversarial machine learning, as it allows adversarial textures to be optimized directly on 3D objects under varying viewpoints and illumination conditions (Athalye et al., 2018; Zhou et al., 2024; Li et al., 2024; Jia et al., 2025). However, a significant limitation persists: differentiable renderers like Mitsuba and Pytorch3D do not support simulation features or complex weather and often require specialized knowledge and manual scene configuration. Thus, adversarial researchers prefer using non-differentiable platforms that offer simulation capabilities like Unreal Engine or CARLA instead (Hull et al., 2024).

These limitations have resulted in a persistent dichotomy within the field leading researchers to choose between:

1. **High-fidelity, non-differentiable simulation**, which offers realism but requires gradient approximation techniques (Xu et al., 2024), or

2. **Differentiable, limited rendering**, which is restricted in its ability to create photorealistic simulations and animations and requires specialized knowledge and manual scene configuration (Laine et al., 2020; Kato et al., 2017; Loper & Black, 2014).

This work aims to address this research gap by introducing a framework that bridges differentiable optimization and high-fidelity 3D environments.

## 3 METHOD

One of the core contributions of UnDREAM is its ability to bridge a longstanding gap between high-fidelity simulation and differentiable rendering — two critical components required for end-to-end optimization in simulation-based environments. Simulation platforms commonly used in adversarial machine learning research provide highly realistic visual outputs and complex scene dynamics, but lack native support for gradient-based optimization. In contrast, differentiable rendering frameworks enable precise, gradient-based optimization but are limited in their ability to generate complex, photorealistic environments with dynamic elements.

**Selection of simulation platform.** While there is a range of platforms to choose from when it comes to picking the simulations, most of the widely used simulation platforms are built on Unreal Engine (Dosovitskiy et al., 2017; Meier et al., 2024b; Shah et al., 2017) making it a natural choice. We implement UnDREAM using Unreal Engine 5 — instead of the Unreal Engine 4 used in the previously mentioned platforms — as it introduces new light scattering techniques such as Nanite and Lumen that improve photorealism in rendering (Engine).

**Selection of differentiable renderer.** Mitsuba is a popular differentiable renderer used in adversarial attacks (Hull et al., 2024). We use Mitsuba in UnDREAM implementation as it a has Python API which allows for integration with existing adversarial ML implementations. Mitsuba, due to its differentiable ray tracing method, has increased photorealism compared to alternatives such as PyTorch3D (Ravi et al., 2020).

UnDREAM unifies the strengths of these two paradigms by combining the visual realism and scene richness of Unreal Engine with the differentiability and optimization capabilities of Mitsuba. This enables gradient-based adversarial optimization directly within photorealistic simulations. More-

Figure 3: The end-to-end UnDREAM optimization pipeline. In each iteration, the adversarial texture is applied to the object in simulation, model predictions are computed, loss is calculated and the gradients are propagated back through the object in the simulation equivalent XML scene to optimize the adversarial texture.

over, UnDREAM is implemented entirely in Python, ensuring seamless compatibility with widely adopted adversarial machine learning frameworks, particularly those built on PyTorch.

## 3.1 SETUP

### 3.1.1 FORWARD RENDERING: SEQUENCES IN UNREAL ENGINE

A LevelSequence is Unreal Engine's native tool for constructing cinematics and is used to capture dynamic scenes with precise temporal and spatial coherence. Users may either import an existing LevelSequence into their Unreal Engine project or create a new sequence from scratch. When creating a new sequence, converting key scene elements such as cameras and dynamic objects into spawnable assets ensures accurate tracking of object motion and reliable reproduction of scenes across frames. For detailed information on creating sequences and their corresponding assets, refer to Appendix B.

### 3.1.2 INVERSE RENDERING: SCENE TRANSFORMATION

Transforming scenes from Unreal Engine to Mitsuba involves reconciling differences across several axes. UnDREAM converts *left-handed Z-up* coordinate system, with *Y-facing* cameras, *Euler* rotations, and *meter* units from Unreal Engine into *right-handed Y-up* coordinate system, with *Z-facing* cameras, *axis-based* rotations, and *centimeter* units in Mitsuba, preserving consistent object–camera geometry across renderings. This transformation process is fully automated and can be executed via the *"initialize.py"* script provided in the project repository. The transformation pipeline involves the following key steps:

1. **Extract camera and object poses.** Retrieve information from the LevelSequence including the 3D positions and orientations of adversarial object, camera, and any other object we would like to track across the frames.

2. **Apply scale and coordinate conversion.** Transform retrieved location and rotation from the LevelSequence from the Unreal Engine coordinate system to the Mitsuba coordinate system and convert the relevant units. This includes adjustments to axes, handedness, and rotation conventions.

3. **Generate Mitsuba XML scenes.** For each frame in the sequence, using the transformed object positions, orientations, and camera parameters create XML files that replicates the

relationship between the camera and adversarial object such that the object appears exactly as it does in the simulation.

4. **Add adversarial texture to XML.** Restore the desired adversarial texture in the XML files in preparation for gradient-based optimization of adversarial texture.

Another intermediate step is replacing the object's texture with a plain white texture in the XML scenes, and save the resulting rendering. This can help us verify XML rendering. Further details about the transformation can be found in Appendix A

## 3.2 RUN ADVERSARIAL ATTACK

While existing attack libraries support a range of gradient-based attacks, their implementations present several limitations in the context of differentiable simulation (Nicolae et al., 2018): (1) they do not support integration with differentiable renderers, (2) they lack mechanisms for inserting updated textures into a simulation at each optimization step, and (3) they do not allow dynamically updating input images based on the output of new renderings after each iteration. UnDREAM enables smooth adaptation of attacks as shown in Fig. 3 with the Algorithm 1.

The attack implementation in UnDREAM builds upon existing adversarial attacks available in widely used attack libraries such as ART (Nicolae et al., 2018), with modifications to enable integration with the differentiable rendering process, as detailed in bridging step of Algorithm 1. Our pipeline iteratively renders scenes, computes gradients for differentiable rendering, and updates textures within simulation. Integrating new attacks into the UnDREAM pipeline requires modifying only **a single line of code** — the line responsible for updating the adversarial texture using the computed gradient in the differentiable renderer as shown in Line 9. By altering this line, users can easily switch between different attack methods supported by existing attack libraries while retaining full compatibility with the differentiable simulation workflow.

---

**Algorithm 1** UnDREAM Attack Pipeline Generates Adversarial Texture by *Bridging* Unreal Simulation and Differentiable Rendering

---

**Input:** input scene $S$ (LevelSequence), victim model $M$, attacker-chosen output $y$, initial texture $T_i$, attack iterations $j$ (equal to number of rendering jobs)
**Ensure:** Adversarial texture $T_{adv}$
1: **Initialize:** $T_{adv} \leftarrow T_i$
2: **for** attack iteration $j$ **do**
3:      $x_{i-n} \leftarrow$ Images from input scene $S$
4:      **procedure** RENDERING JOB FINISH CALLBACK
5:          Get model predictions $pred \leftarrow M(x)$
6:          Find $L \leftarrow Loss(y, pred)$
7:          Calculate gradients $g \leftarrow \nabla L$
8:          **Pass gradients to differentiable renderer**          ▷ Bridging step
9:          *Using differentiable renderer:* Use gradients to update texture
10:         Save updated $T_{adv}$
11:         Update new $T_{adv}$ as object texture in Unreal Engine
12:      **end procedure**
13: **end for**

---

## 4 EXPERIMENTS

We evaluate the capabilities of UnDREAM along two primary axes: the level of control afforded within the simulation environment, and the flexibility and effectiveness of adversarial attacks implemented through the framework. These two dimensions highlight UnDREAM's utility both as a research tool for simulation-based adversarial machine learning and as a platform for conducting precise, grounded attacks. An overview of the available controls spanning both simulation and optimization components is presented in Table 2. This includes parameters such as scenes, objects, lighting, models, tasks and algorithms.

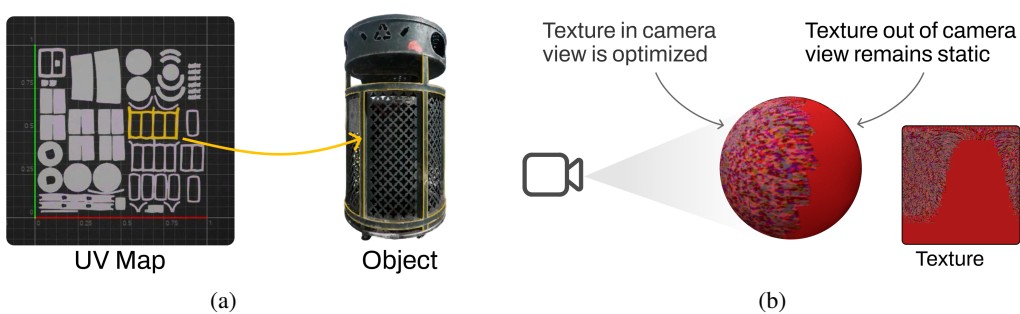

UV Map      Object

Texture in camera view is optimized

Texture out of camera view remains static

Texture

(a)              (b)

Figure 4: **(a)** UV layout and object mesh in the UV editor display in Unreal Engine 5. Each segment from the UV map corresponds to a part on the object. **(b)** Optimized object (sphere) shown from the side. Only the part of the texture that is visible in the camera view is optimized.

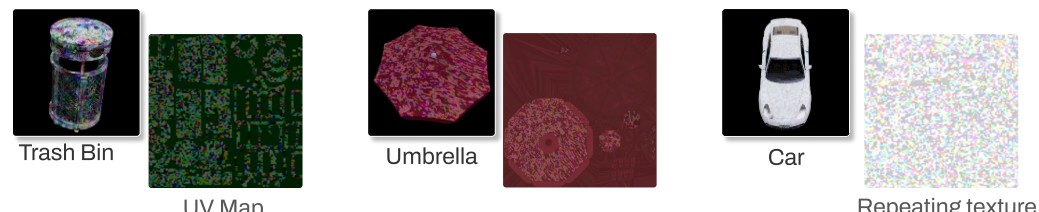

Trash Bin         Umbrella         Car

UV Map                                                     Repeating texture

Figure 5: UnDREAM enables optimization of 3D objects of arbitrary shapes. If the object is very large (eg: car), Unreal allows for the texture to repeat, thus covering the whole object.

## 4.1 OBJECT

One of the main contributions of UnDREAM is it enables researchers to optimize adversarial textures for any arbitrary 3D object in high-fidelity simulations. This will open many new research directions as the current optimization pipeline uses 2D approximation techniques when dealing with photorealistic simulations (Xu et al., 2024), thus limiting the adversarial objects possible to only 2D rectangular shapes. Objects in the simulation utilize a technique known as UV mapping to project a 2D texture onto their 3D surface. This mapping allows us to determine how each pixel in the texture corresponds to a specific point on the object's surface, facilitating precise gradient-based updates during optimization. An example of the UV map editor in Unreal Engine is shown in Fig. 4a. Upon completion of the optimization process, we observe that only the regions of the texture directly visible to the camera are significantly modified, as illustrated in Fig. 4b. Fig. 5 presents several examples of objects used in our experiments, along with their optimized adversarial textures.

## 4.2 ENVIRONMENT AND LIGHTING

Environmental and lighting conditions play a critical role in adversarial machine learning, particularly in the context of physically realizable attacks. Variations in lighting, shadows, time of day, or weather can significantly influence the effectiveness and transferability of adversarial attacks (Wei et al., 2024). However, existing datasets and simulation pipelines typically offer limited support for systematically varying environmental parameters. Even when present, they are often not available as controllable inputs for experimentation, but preset variables (Nesti et al., 2024). In contrast, UnDREAM provides fine-grained control over lighting and environmental conditions within the simulation, enabling researchers to evaluate the robustness of adversarial attacks across diverse settings. Fig. 6 demonstrates examples of the same scene rendered under different lighting and weather conditions using UnDREAM.

## 4.3 ATTACKS AND TASKS

UnDREAM enables seamless experimentation with a wide range of adversarial attacks and tasks within a high-fidelity simulation environment. As described in Algorithm 1, the structure of the at-

| Element | | Possibilities | Demonstrated |
|---|---|---|---|
| Simulation | Objects | **Any** object (.OBJ) | sphere, bin, car, umbrella |
| | Scenes | **Any** scene | park, city |
| | Lighting | **Any** lighting or weather variation | sunny, cloudy, dark, rainy |
| Attacks | Models | **Any** model from HuggingFace | detr-resnet-50 |
| | Tasks | Can be expanded to **any** tasks | object detection, classification |
| | Algorithms | Can be expanded to **any** attack | PGD, Auto-PGD |

Table 2: Overview of configurable elements supported by UnDREAM, including simulation components and adversarial attack settings.

| Attack | Budget | Benign | | Adversarial | |
|---|---|---|---|---|---|
| | | Accuracy (%) (Cls) | mAP (Det) | Accuracy (%) (Cls) | mAP (Det) |
| PGD | 0.78 | 100 | 100 | 23.8 | 29.5 |
| | 1.00 | | | 9.5 | 14.3 |
| Auto-PGD | 0.78 | 100 | 100 | 28.5 | 33.7 |
| | 1.00 | | | 9.5 | 6.9 |

Table 3: Attack results for attacks in UnDREAM with classification (Cls) and detection (Det) tasks. Results are recorded as accuracy for classification and mean average precision for detection

tack implementation in UnDREAM mirrors that of popular adversarial attack libraries. This ensures that attacks can be easily adapted and integrated into the simulation with minimal code modifications. We provide code implementation for attacking models detr-resnet-50, yolov8, yolo v11 on tasks such as image classification and object detection, available in the code repository. As examples to demonstrate successful attack execution, Table 3 shows the results for the PGD and Auto-PGD attacks on detr-resnet-50 for the tasks classification and detection of a person walking in a park; reducing the classification accuracy from 100% to 9.5%, and detection mAP from 100% to 6.9% a significant drop commonly seen in unbounded attacks (Xu et al., 2024).

## 5 LIMITATIONS AND FUTURE WORK

UnDREAM represents the first framework of its kind, offering a unique integration of high-fidelity simulation with differentiable rendering, and opening up several promising research directions for the adversarial machine learning community. However, there remains significant room for further development and enhancement. Currently, UnDREAM supports a wide variety of scenes, sequences, camera angles, object trajectories, 3D models, and lighting configurations. Nonetheless, there is potential to expand this collection into a more comprehensive and diverse set of ready to use simulation scenarios for evaluation. Achieving this will require careful curation to ensure coverage across a broad spectrum of parameters including object geometry, texture complexity, environmental conditions, and camera perspectives. It is also essential that these scenes and assets remain freely and easily accessible to the research community, enabling reproducibility and broad adoption. Beyond expanding the simulation assets, our future work includes incorporating a broader range of attack methods into UnDREAM, allowing users to experiment with and benchmark diverse adversarial strategies. UnDREAM can also be extended to evaluate defensive techniques, creating a unified environment where both attacks and defenses can be tested under realistic, high-fidelity conditions. In this way, UnDREAM closes the gap between synthetic evaluations and physical-world deployment, further strengthening research on adversarial robustness.

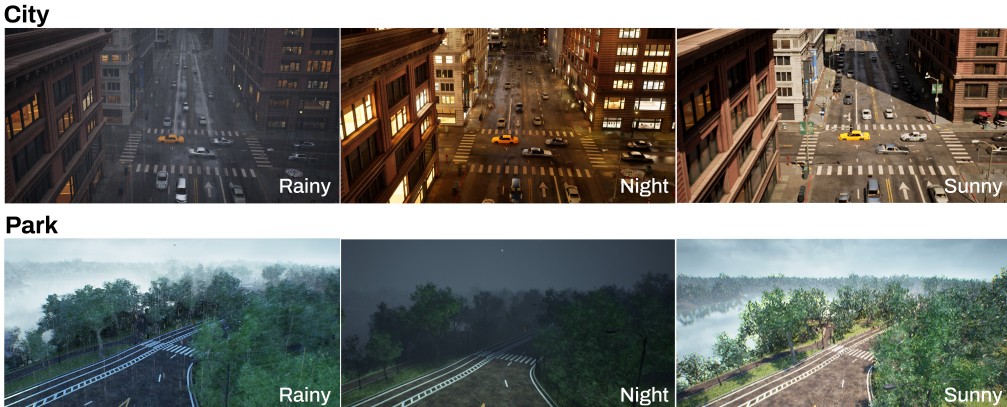

Figure 6: The same scene rendered under different lighting and weather conditions using UnDREAM demonstrating its ability to replicate environmental factors. A city scene (top) and a park scene (bottom) is rendered in rainy, night and sunny environments.

## 6 CONCLUSION

In this work we introduce UnDREAM: a framework that bridges the gap between high-fidelity simulation environments and differentiable optimization, combining the realism of photorealistic simulators with the flexibility of gradient-based methods. UnDREAM enables the transformation of 3D objects and textures from Unreal Engine into a fully differentiable system, enabling end-to-end optimization of adversarial patterns on arbitrarily shaped 3D objects within realistic environments. In contrast to traditional approaches that optimize textures externally and later superimpose them onto pre-rendered simulation frames, UnDREAM performs optimization directly within the simulation environment. This allows adversarial textures to be optimized as they naturally appear in the scene, preserving critical factors such as lighting, and material properties. As a result, UnDREAM supports the optimization of more complex and realistic adversarial perturbations beyond simple 2D patches. By embedding adversarial textures directly into the simulation pipeline, UnDREAM removes the need for manual specification of patch boundaries, coordinate transformations, or enforcing consistency across video frames. This leads to more coherent behavior across consecutive frames which is crucial for evaluating adversarial robustness in tasks such as object tracking, where temporal consistency is essential. Thus, UnDREAM enables more faithful assessments of both adversarial attacks and the defenses designed to counter them. UnDREAM provides a flexible platform to test and compare attack strategies across diverse conditions. It supports extensive customization of environments, objects, weather, and lighting, allowing researchers to craft tailored scenarios for evaluating adversarial performance and robustness.

## 7 REPRODUCIBILITY STATEMENT

We work to ensure reproducibility of the paper and its results. We include code along with the assets we developed for testing such as LevelSequences Appendix B, and Niagara Systems for rain § C.1 in the anonymized repository https://anonymous.4open.science/r/UnDREAM. We have also included detailed instructions to replicate our results in § 3 and Appendix A. We have also included downloadable links for hte free assets used that are sourced from the Unreal Fab Store Appendix C.

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

## A  TRANSFORMATION DETAILS

Transformation of a scene from Unreal Engine to Mitsuba is the first fundamental step in bridging simulation with a differentiable renderer. The difference between these platforms are detailed in Table 4.

| Parameter | Unreal Engine | Mitsuba |
|---|---|---|
| World Coordinate | Left handed Z up | Right handed Y up |
| Camera orientation | Facing Y | Facing Z |
| | Z up | Y up |
| Rotation | XYZ Euler | Coordinate axes |
| Scale | meters | centimeters |

Table 4: Comparison of coordinate systems and transformation conventions between Unreal Engine and Mitsuba.

While scale conversion is relatively easy, the steps to tackle coordinate and rotation conversion are expanded on here:

**World coordinate:**  To convert Left handed Z up to a Right handed Y up system, we must swap values on the Y and Z axes. Keep in mind this also affects rotation, as the rotation for Mitsuba is defined by coordinate axes. The transformation can be understood as Eq. 1. Another way to convert between the systems is rotating the Mitsuba frame by 90 degrees around x, and y by 90 degrees and negating the y axis.

$$\begin{bmatrix} 1 & 0 & 0 \\ 0 & 0 & 1 \\ 0 & 1 & 0 \end{bmatrix} \tag{1}$$

**Rotation:**  To account for the rotation between XYZ Euler and coordinate axes, needs to be done very carefully. Unless the pitch (rotation around Y) in Unreal Engine is a $\pm$ 90 degrees, we can simply rotate the camera and object at the origin before translating it to the desired position. However, if the Y rotation is $\pm$ 90 degrees, the object is subject to Gimbal Lock, a phenomenon in which the object loses on of its degrees of freedom, thus rotations around Z axis, may not actually apply.

## B  RENDERING IMAGES

### B.1  CREATE SEQUENCES

Creating custom sequences in Unreal Engine is a straightforward process that enables generation of cinematic videos. In the given code repository, the "initialize.py" script is designed to recognize and extract the 3D positions and orientations of various objects, including the camera, adversarial objects, and any other elements specified for tracking across frames. To enable this functionality, it is essential to define "keyframes" for all relevant objects, indicating their positions and orientations at specific points in the sequence such as start, end and any other points of trajectory change. We rely on these keyframes to interpolate the intermediate poses of each object throughout the sequence. Unreal Engine employs a similar interpolation mechanism, computing object transformations between keyframes to ensure smooth transitions and continuity in motion.

### B.2  ADD ANIMATION

In Unreal Engine, animation refers to the movement of a skeletal mesh associated with a particular object. These animations enable the creation of fluid and realistic motion within a sequence, enhancing the overall visual quality. While it is possible to design and implement custom animations for any skeletal mesh, doing so is not always necessary. A wide range of pre-animated skeletal meshes

is available through the Fab Store (also called Unreal Marketplace), where users can download high-quality assets for use in their projects. Examples of animations for a person can be different iterations of walking, jogging, running, climbing up or down stairs, sitting down, standing up, etc. Utilizing multiple of these together can help us create realistic videos. We can also have complex camera movements very easily which gives sweeping cinematic effects to generated videos.

## C  CHANGING ENVIRONMENT

One of the key advantages Unreal Engine offers over other simulation frameworks is its ability to support the creation of highly diverse and customizable environments. In this work, two environments were used: CitySample [1] (an urban setting) , and CityParkEnvironment [2] (a park scene). Both assets are available for free on the Unreal Fab Store and their licensing is permissible for research applications. While additional environments can certainly be utilized, it is important to note that increased environmental complexity typically leads to higher computational demands during the optimization cycle.

### C.1  NIAGARA SYSTEM

The Niagara system in Unreal Engine provides a powerful framework for creating a wide range of visual effects within a 3D environment. This includes environmental phenomena such as rain, snow, fog, fire, fountains, and other effects that improve realism and diversity of scenes. In this work, Niagara was used to simulate environmental effects, examples of which can be seen in Fig. 1 and Fig. 6. The system offers extensive configuration abilities through its user interface, allowing for precise control over particle behavior, spawn rate, lifetime, velocity, trailing ribbon for streaks, etc. The Niagara UI for the rain effect used in this study is illustrated in Fig. 7.

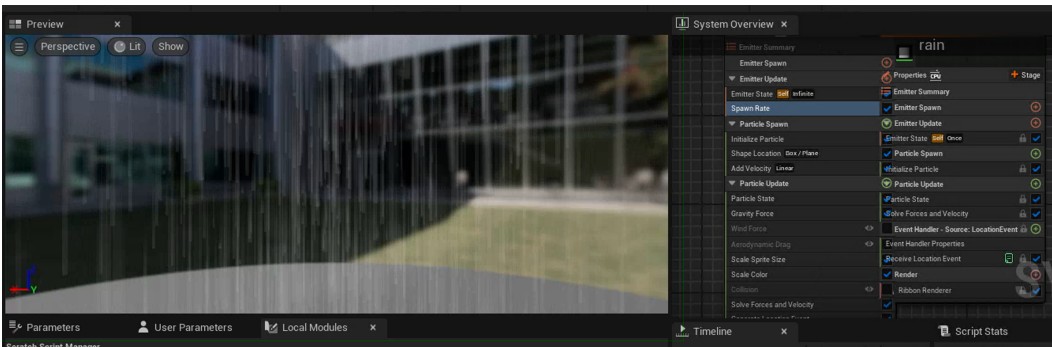

Figure 7: Niagara system interface used to create and configure the rain effect in the environment.

---

[1]https://www.fab.com/listings/4898e707-7855-404b-af0e-a505ee690e68

[2]https://www.fab.com/listings/11cc2abb-126c-4452-9fe4-6f2381d96544

