# OpenReview forum: "UnDREAM: Bridging Differentiable Rendering and Photorealistic Simulation for End-to-end Adversarial Attacks"
_ICLR.cc/2026/Conference — ICLR 2026 Conference Withdrawn Submission_

### Official Review · Reviewer_CHr5 · 2025-10-28

**Soundness:** 3
**Presentation:** 3
**Contribution:** 2
**Rating:** 4
**Confidence:** 4

**Summary:**

Paper presents UnDREAM, a framework that connects photorealistic simulators (Unreal Engine 5) with differentiable renderers (Mitsuba) to enable gradient-based adversarial optimization directly inside realistic environments. It automates scene conversion, synchronization of geometry and lighting, and integrates standard attacks (PGD, Auto-PGD) for optimizing textures on 3D objects. The framework primarily contributes an engineering bridge that allows differentiable rendering within simulation, expanding experimental possibilities rather than introducing a new algorithm.

**Strengths:**

Addresses a real gap between differentiable rendering and simulation.

Technically sound framework with automatic Unreal to Mitsuba conversion.

Enables new realistic adversarial-attack studies with full control of lighting and motion.

Good documentation and reproducibility; code publicly available.

Clear and visually strong presentation.

**Weaknesses:**

Conceptual novelty is minimal, and largely an engineering integration.

Experimental evaluation is shallow: few attacks, one detector model, no baselines or runtime data.

No demonstration of transferability to real-world settings or defenses.

‘First-of-its-kind’ claim is slightly overstated given prior differentiable-simulation works (e.g., RenderBender 2024).

**Questions:**

What is the runtime overhead per iteration compared to using only Mitsuba or Unreal?

How does UnDREAM handle dynamic effects (rain, shadows) during gradient backpropagation?

Can attacks optimized with UnDREAM transfer to real or physical scenes?

---

### Official Review · Reviewer_ZiLA · 2025-10-29

**Soundness:** 2
**Presentation:** 2
**Contribution:** 3
**Rating:** 2
**Confidence:** 4

**Summary:**

The paper introduces UnDream, a software framework for testing autonomous systems that combines photorealistic simulations via UnrealEngine with differentiable rendering via Mitsuba.
Through this combination it becomes possible to adversarially optimize object textures in otherwise non-differentiable simulation environments, and perform physical texture-based adversarial attacks.
As a result, the adversarial texture is embedded with a high degree of realism and will be rendered faithfully under diverse weather and lighting conditions.

**Strengths:**

The proposed software framework is **original** in itself, and to the best of my knowledge, the first solution that combines complex established simulation environments and differentiable rendering at such a deep level.
The **quality** of the contributions is mixed. While the provision of an open-source software framework is a high-quality contribution, the evaluation lacks rigor and depth (see weaknesses).
In terms of **clarity**, I found the paper generally well written and clear, though some minor questions about the framework execution remain (see questions) and the related work on differentiable rendering and adversarial attacks would profit from a broader perspective on the topic (see weaknesses).
In my view, its **significance** for the robustness research community is the biggest strength of this paper. The proposed system is very valuable for future exploration of realistic adversarial attacks, and was purposefully designed to be easy to use with its open-source implementation and the minimal changes required to add new attacks.

**Weaknesses:**

In its current stage, the lack of experiments is the biggest weakness of this paper. Only one quantitative experiment is provided, and comparisons to other attack frameworks are entirely missing.
* Even though the paper makes it clear that it is easy to add adversarial attacks, only two (PGD and AutoPGD) are demonstrated.
* There are no comparisons to prior physical attack systems
* The provided quantitative assessment remains high-level, and individual evaluation results for det-resnet-50, yolov8, yolov11 are not reported in the main paper or the appendix. Also, only evaluating the attack performance on three models appears very reduced, and reasons for this slim evaluation are not given.
* I could not find experimental evidence for the second claimed contribution (this new system provides more faithful evaluation of both adversarial attacks and corresponding defenses). Defenses were not evaluated.

Regarding related work, the contribution would profit from being embedded more broadly in the physical attack literature (not autonomous driving only). For example, earlier works on adversarial weather have started to bridge the gap between realistic physical simulations and adversarial attacks [1,2,3]. Without more discussion, the claim that UnDream is the first software framework to bridge the gap between photorealistic simulators and differentiable rendering (L.045) appears too strong.

[1] Schmalfuss, Mehl, Bruhn. Distracting downpour: Adversarial weather attacks for motion estimation. ICCV'23

[2] Zhong, Liu, Zhai, Jiang, Ji. Shadows can be dangerous: Stealthy and effective physical-world adversarial attack by natural phenomenon. CVPR'22

[3] Gao, Guo, Juefei-Xu, Yu, Feng. AdvHaze: Adversarial haze attack. Arxiv'2021

**Questions:**

* In Sec. 3.2 and the corresponding algorithm, it did not become clear to me how many times the image is rendered via UnrealEngine. Is it only 1x in the beginning and 1x in the end to re-insert the optimized texture, or is it called in every attack iteration?
* Do the adversarial attacks optimize per-image textures, or does the optimization yield textures optimized over a video sequence?
* How does the attack performance of attacks with UnDream compare to attacks that add textures after rendering or include 2D textures only?
* How does the performance of det-resnet-50, yolov8, yolov11 and other models vary under the different adversarial attacks (like PGD and AutoPGD)?

---

### Official Review · Reviewer_snDM · 2025-10-30

**Soundness:** 2
**Presentation:** 2
**Contribution:** 1
**Rating:** 2
**Confidence:** 4

**Summary:**

This paper introduces UnDREAM, a program that builds an adversarial attack pipeline in Unreal Engine. This system enables end-to-end differentiable optimization that manipulates adversarial texture mapping directly in Unreal Engine 5, thanks to differentiable rendering (using Mitsuba in this paper).

**Strengths:**

It is important to bridge the gap between photo-realistic rendering pipelines and adversarial attack pipelines that break trained models via manipulations of texture mapping.

**Weaknesses:**

This paper is too superficial in its writing and in verifying the effectiveness of its adversarial pipeline. Some issues are:

* The header is “Under review as a conference paper at ICLR 2025,” which implies a potential misuse of the template.

* It does not explain how L307–308 (L7–8 in A1) are executed end to end in the UnDREAM pipeline. Adversarial attacks require gradient ascent or another specific gradient-update design, which is not described in this paper. The paper provides examples with PGD and Auto-PGD, but it is unclear to readers how to adopt these attack methods. It is also unclear whether the gradient is back-propagated with respect to the entire patch/UV atlas/texture coordinates. Moreover, whether occlusion in rendering and distortion of texture affect the performance of the adversarial attack is not discussed. These are well-known issues, but they appear to be ignored here, and the gradient of unseen areas seems to be implicitly assumed to be 0.

* Table 3 only shows results for DETR-ResNet-50, but L409 indicates YOLOv8 and YOLOv11 are also implemented. Please provide results for these victim models as well. Even if UnDREAM is assumed to be effective, there is no figure or visualization showing a successful attack scenario. Moreover, if there is only one scenario—“a person walking in a park” (L412)—what evidence supports the claim that UnDREAM is versatile and can generalize to all configurable scenes discussed in the paper and in Table 2?

* There are no visual examples demonstrating effectiveness under different weather conditions (e.g., Figure 6 does not present results indicating successful attacks in those environments, which is confusing).

* The paper does not cite [DR.JIT](https://dl.acm.org/doi/abs/10.1145/3528223.3530099), even though the differentiable rendering pipeline in the implementation relies on it.

**Questions:**

NA. See weakness.

---

### Official Review · Reviewer_CqX4 · 2025-11-01

**Soundness:** 1
**Presentation:** 2
**Contribution:** 3
**Rating:** 4
**Confidence:** 4

**Summary:**

This paper proposes an open-source software framework named UnDREAM (Differentiable Rendering for End-to-End Adversarial Modeling), which integrates a photorealistic simulator (Unreal Engine 5) with a physically based differentiable renderer (Mitsuba) to enable end-to-end optimization of adversarial attacks in realistic 3D environments.

The paper highlights the existing trade-off between using non-differentiable but highly photorealistic simulators and differentiable but less realistic renderers that typically lack support for complex parameters and easy scene manipulation.

UnDREAM addresses this by rendering image sequences in Unreal Engine, mirroring the scene setup in Mitsuba through precise camera and object pose conversion, backpropagating gradients through the differentiable renderer, and then re-inserting the optimized texture into Unreal for evaluation.

The methodology covers platform and renderer selection, setup for both forward and inverse rendering, and the full adversarial optimization pipeline. Experiments demonstrate the system’s configurability (e.g., object, scene, and lighting settings) and flexibility across attack types, including PGD and Auto-PGD on both classification and object detection tasks.

**Strengths:**

1. **Clear problem and solution:** The paper clearly identifies a key limitation in existing pipelines. Non-differentiable simulators require “optimize-then-paste” approaches that fail to capture lighting, material, and pose interactions. UnDREAM addresses this gap by enabling end-to-end differentiable optimization within a photorealistic environment.

2. **Practical and technically sound pipeline:** The integration between Unreal Engine 5 and Mitsuba through a render-callback loop for texture updates represents strong engineering work. This bridge combines the strengths of both simulators, providing Unreal’s realism together with Mitsuba’s differentiability, and offers a practical foundation for future research.

3. **Open-source and reproducibility commitment:** The authors provide an anonymized repository with detailed setup instructions and asset links. This openness improves reproducibility and is likely to have a positive impact on the research community working on physically grounded adversarial attacks.

**Weaknesses:**

1. **Missing citation and potentially misleading claim:** The paper strongly claims that their proposal is the first software framework that bridges the gap between photorealistic simulators and differentiable renderers to enable end-to-end optimization of adversarial attacks. This problem and solution have actually been raised and addressed by [1], which proposes a neural renderer (called DTN) to transform adversarial texture into realistic rendered images based on reference images from CARLA/Unreal Engine. The work is then extended by [2], [3], [4], which also combine differentiable renderers and photorealistic simulators (CARLA). The claim may still hold if framed as the first open-source bridge between Unreal Engine 5 and a general-purpose differentiable renderer (Mitsuba), but the current “first” statement is too broad. It should be narrowed and supported by citations to related works that have already explored similar ideas.

2. **Lighting/material fidelity in the bridge is under-specified:** Section 3.1.2 details geometry and camera transforms but does not explain how materials/BRDFs, lighting (Lumen), exposure/tonemapping, or shadowing are mapped or approximated between UE and Mitsuba. Without consistent light/material transfer, this would be similar to optimizing the texture without considering accurate lighting and material as in the “optimize-then-paste” related pipelines; as a result, gradients may optimize for a different rendering model than the simulator.

3. **Limited attack evaluation and baselines:**
- Table 3 shows large drops under PGD/Auto-PGD but the budgets (0.78, 1.00) are not clearly defined (norm? per-pixel? texture parameterization?).
- The evaluation is limited to two adversarial scenarios (classification and detection with two attack methods) without explanation or visualization of which target objects are being attacked.
- Comparisons to standard pipelines (optimize-outside-sim then paste-in) are missing.
- No comparison to attacks using differentiable renderers alone (e.g., PyTorch3D or Mitsuba-only) is provided to quantify the benefit of bridging the simulator.

4. **Lack of clarity in the end-to-end optimization pipeline (Fig. 3):** Referring to Fig. 3, the optimization pipeline appears to be: Texture -> Unreal (non-differentiable) -> Rendered Image -> Model -> Loss & Gradient Calculation -> Pass gradients to Mitsuba (differentiable renderer) -> Texture. Let texture is $t$, Unreal is $U(...)$, Rendered Image is $x$, Model is $M(...)$, and Loss is $L(...)$, we can formulaize that $Optimization Loss = L(M(x))$ where $x = U(t, ...)$. If $U$ is not differentiable, how can we obtain the gradient of the loss with respect to the texture $t$?

5. **No evaluation between Mitsuba renders and Unreal renders:** Since gradients are calculated using the differentiable renderer, there should be quantitative and qualitative evaluation of how accurate the mirrored Mitsuba render is compared to the ground-truth scene from Unreal Engine. Suggested metrics include PSNR, SSIM, LPIPS, or other relevant measures.

---

References

[1] Naufal Suryanto, Yongsu Kim, Hyoeun Kang, Harashta Tatimma Larasati, Youngyeo Yun, Thi-Thu-Huong Le, Hunmin Yang, Se-Yoon Oh, Howon Kim; Proceedings of the IEEE/CVF Conference on Computer Vision and Pattern Recognition (CVPR), 2022, pp. 15305-15314
[2] Naufal Suryanto, Yongsu Kim, Harashta Tatimma Larasati, Hyoeun Kang, Thi-Thu-Huong Le, Yoonyoung Hong, Hunmin Yang, Se-Yoon Oh, and Howon Kim. ACTIVE: Towards highly transferable 3D physical camouflage for universal and robust vehicle evasion. In Proceedings of the IEEE/CVF International Conference on Computer Vision, pp. 4305–4314, 2023.
[3] Jiawei Zhou, Linye Lyu, Daojing He, and Yu Li. RAUCA: a novel physical adversarial attack on vehicle detectors via robust and accurate camouflage generation. In Proceedings of the 41st International Conference on Machine Learning, pp. 62076–62087, 2024.
[4] Linye Lyu, Jiawei Zhou, Daojing He, and Yu Li. CNCA: Toward customizable and natural generation of adversarial camouflage for vehicle detectors. In The Thirty-eighth Annual Conference on Neural Information Processing Systems (NeurIPS 2024), 2024.

**Questions:**

Please check and clarify the issues raised in Points 2, 3, and 4 of the weaknesses section.

---

### Note · Authors · 2025-11-18

I have read and agree with the venue's withdrawal policy on behalf of myself and my co-authors.